# Deep Bayesian Structure Networks

## Abstract

Bayesian neural networks (BNNs) introduce uncertainty estimation to deep networks by performing Bayesian inference on network weights. However, such models bring the challenges of inference, and further BNNs with weight uncertainty rarely achieve superior performance to standard models. In this paper, we investigate a new line of Bayesian deep learning by performing Bayesian reasoning on the structure of deep neural networks. Drawing inspiration from the neural architecture search, we define the network structure as gating weights on the redundant operations between computational nodes, and apply stochastic variational inference techniques to learn the structure distributions of networks. Empirically, the proposed method substantially surpasses the advanced deep neural networks across a range of classification and segmentation tasks. More importantly, our approach also preserves benefits of Bayesian principles, producing improved uncertainty estimation than the strong baselines including MC dropout and variational BNNs algorithms (e.g. noisy EK-FAC).

## 1 Introduction

Bayesian deep learning aims at equipping the flexible and expressive deep neural networks with appropriate uncertainty quantification (MacKay, 1992; Neal, 1995; Hinton & Van Camp, 1993; Graves, 2011; Blundell et al., 2015; Gal & Ghahramani, 2016). Traditionally, Bayesian neural networks (BNNs) introduce uncertainty in the network weights, addressing the over-fitting issue which standard neural networks (NNs) are prone to. Besides, the predictive uncertainty derived from the weight uncertainty is also of central importance in practical applications, e.g., medical analysis, automatic driving, and financial tasks.

Modeling the uncertainty on network weights is plausible and well-evaluated (Blundell et al., 2015; Ghosh et al., 2018). However, BNNs usually preserve benefits of Bayesian principles such as well-calibrated predictions at the expense of compromising performance and hence are impractical in real-world applications (Osawa et al., 2019), due to various reasons. On one hand, specifying a sensible prior for networks weights is difficult (Sun et al., 2019; Pearce et al., 2019). On the other hand, the flexible variational posterior of BNNs comes with inference challenges (Louizos & Welling, 2017; Zhang et al., 2018; Shi et al., 2018). Recently, the efficient particle-based variational methods (Liu & Wang, 2016) have been developed with promise, but they still suffer from the particle collapsing and degrading issues for BNNs due to the high dimension of the weights and the over-parameterization nature of such models (Zhuo et al., 2018; Wang et al., 2019).

In this work, we investigate a new direction of Bayesian deep learning that performs Bayesian reasoning on the structure of neural networks while keeping the weights as point estimates. We propose an approach, named *Deep Bayesian Structure Networks* (DBSN). Specifically, in the spirit of differentiable neural architecture search (NAS) (Liu et al., 2019; Xie et al., 2019), DBSN builds a deep network by repeatedly stacking a computational cell in which any two nodes (i.e. tensors) are connected by redundant transformations (see Figure 1). The network structure is defined as the gating weights on these transformations, whose distribution is much easier to capture than those of the high-dimensional network weights. To jointly optimize the network weights and the parameterized distribution of the network structure, we adopt a stochastic variational inference paradigm (Blundell et al., 2015) and use the reparameterization trick (Kingma & Welling, 2013). One technical challenge is driving DBSN to achieve satisfying convergence, since the network weights can hardly fit all the structures sampled from the structure distribution. To overcome this challenge, we propose two techniques. First, we advocate reducing the variance of the sampled structures with a simple mod-

Figure 1: BNNs with uncertainty on the weights (left) vs. DBSN with uncertainty on the network structure (right) (we only depict three operations between tensors $\boldsymbol{N}^1$ and $\boldsymbol{N}^2$ for simplicity). $\boldsymbol{w}$ and $\boldsymbol{\alpha}$ represent network weights and network structure, respectively. In DBSN, $\boldsymbol{w}$ is also learnable.

ification of the sampling procedure. Second, we suggest using a more compact structure learning space than that of NAS, to make the training more feasible and more efficient.

There are at least two motivations that make DBSN an appealing choice: 1) DBSN bypasses the frustrating difficulties of characterizing weight uncertainty and enables the performance-enhancing structure learning (Zoph & Le, 2016; Liu et al., 2019), so DBSN shall have better predictive performance than classic BNNs. 2) Previous analysis (Wang et al., 2019) shows that due to the overparametrization nature of BNNs, the state-of-the-art inference algorithms for weight uncertainty can suffer from mode collapsing, as multiple configurations of weights with a fixed structure correspond to one single function. In contrast, DBSN compactly models the uncertainty of structure and performs inference in a much lower-dimensional space, avoiding this issue and hence being able to exhibit more calibrated predictive uncertainty. Moreover, in the perspective of NAS, DBSN is also promising as it provides another principled way to learn network structures by resorting to the Bayesian formalism instead of the widely used meta-learning formalism in differentiable NAS.

To empirically validate these hypotheses, we evaluate DBSN with extensive experiments. We first testify the data fitting and structure learning ability of DBSN on challenging classification and segmentation tasks. Then, we compare the quality of predictive uncertainty estimates via calibration, which is a common concern in the community. We further evaluate the predictive uncertainty on adversarial examples and out-of-distribution samples, drawn from shifted distributions from the training data, to verify whether the model *knows what it knows*. At last, we perform an experiment to validate a promising application of DBSN in the one-shot NAS (Bender et al., 2018; Guo et al., 2019). Surprisingly, across all the tasks, DBSN consistently achieves comparable or even better results than the strong baselines.

## 2 BACKGROUND

We first review the necessary background for DBSN and then elaborate DBSN in the next section.

### 2.1 STOCHASTIC VARIATIONAL INFERENCE FOR BNNS

Let $\mathcal{D} = \{(x_i, y_i)\}_{i=1}^N$ be a set of $N$ data points. BNNs are typically defined by placing a prior $p(\boldsymbol{v})$ on some variables of interest (e.g., network weights or network structure) and the likelihood is $p(\mathcal{D}|\boldsymbol{v})$. Directly inferring the posterior distribution $p(\boldsymbol{v}|\mathcal{D})$ is intractable because it is hard to integrate w.r.t. $\boldsymbol{v}$ exactly. Instead, variational BNNs (Hinton & Van Camp, 1993; Graves, 2011; Blundell et al., 2015) suggest approximating $p(\boldsymbol{v}|\mathcal{D})$ with a $\boldsymbol{\theta}$-parameterized distribution $q(\boldsymbol{v}|\boldsymbol{\theta})$ by minimizing the Kullback-Leibler (KL) divergence between them:

$$\min_{\boldsymbol{\theta}} D_{\mathrm{KL}}(q(\boldsymbol{v}|\boldsymbol{\theta})\|p(\boldsymbol{v}|\mathcal{D})) = -\mathbb{E}_{q(\boldsymbol{v}|\boldsymbol{\theta})}[\log p(\mathcal{D}|\boldsymbol{v})] + D_{\mathrm{KL}}(q(\boldsymbol{v}|\boldsymbol{\theta})\|p(\boldsymbol{v})) + \log p(\mathcal{D}), \quad (1)$$

where $\log p(\mathcal{D})$ is a constant w.r.t. $\boldsymbol{\theta}$ and usually omitted in the minimization. To solve problem (1), the most commonly used method is the low-variance reparameterization trick (Kingma & Welling, 2013; Blundell et al., 2015), which replaces the sampling procedure $\boldsymbol{v} \sim q(\boldsymbol{v}|\boldsymbol{\theta})$ with the corresponding deterministic transformation $\boldsymbol{v} = t(\boldsymbol{\theta}, \epsilon)$ with a sample of parameter-free noise $\epsilon$, to enable the direct gradient back-propagation through $\boldsymbol{\theta}$.

### 2.2 CELL-BASED DIFFERENTIABLE NEURAL ARCHITECTURE SEARCH (NAS)

Cell-based NAS has shown promise (Zoph et al., 2018; Pham et al., 2018) and been developed to be differentiable for better scalability (Liu et al., 2019; Xie et al., 2019; Weng et al., 2019).

Generally, the network in cell-based differentiable NAS[1] is composed of a sequence of cells (e.g., modules) which have the same internal structure and are separated by upsampling or downsampling modules. Every cell contains $B$ sequential nodes (i.e., tensors): $N^1, \ldots, N^B$. Each node $N^j$ is connected to all of its predecessors $N^i$ so long as $i < j$ by $K$ possible redundant operations $o_1^{(i,j)}, \ldots, o_K^{(i,j)}$, e.g., convolution, skip connection, pooling. The network structure is defined as $\alpha = \{\alpha^{(i,j)} | 1 \le i < j \le B\}$ where $\alpha^{(i,j)} \in \Delta^{K-1}$ corresponds to the gating weights on the $K$ available operations from $N^i$ to $N^j$. Therefore, the information gathered from $N^i$ to $N^j$ is a weighted sum of the outputs from $K$ different operations on $N^i$ (we denote the set including the parameters of all the operations in the network as $w$):

$$N^{(i,j)} = \sum_{k=1}^{K} \alpha_k^{(i,j)} \cdot o_k^{(i,j)}(N^i; w). \tag{2}$$

Then, the node $N^j$ is calculated by summing all the information from its predecessors:

$$N^j = \sum_{i<j} N^{(i,j)}. \tag{3}$$

Meta-learning-like gradient descent is adopted for optimization to reduce the prohibitive computational cost needed by RL or evolution (Liu et al., 2019; Xie et al., 2019). However, the goal of the optimization is the network structure instead of the model performance. Thus, after training, this kind of NAS needs to prune the searched structure and re-train a new network model with the compact structure for performance comparison, which is labor-intensive and is avoided in our work.

## 3 DEEP BAYESIAN STRUCTURE NETWORKS

In this work, we propose a novel Bayesian structure learning approach for the deep neural networks. Concretely, we follow the network design of NAS but we view $\alpha$ as Bayesian variables and $w$ as point estimates (see the graphical model in Figure 1). To infer the posterior distribution $p(\alpha | \mathcal{D}, w) = \frac{p(\alpha)p(\mathcal{D}|\alpha, w)}{p(\mathcal{D})}$, where $p(\alpha)$ is the prior (we omit its dependency on the hyperparameter $\theta_0$ here), we adopt the techniques in Section 2.1. We assume both the prior and the introduced variational are fully factorizable categorical distributions, namely, $p(\alpha) = \prod_{i<j} p(\alpha^{(i,j)})$ and $q(\alpha|\theta) = \prod_{i<j} q(\alpha^{(i,j)}|\theta^{(i,j)})$, where $\theta = \{\theta^{(i,j)} \in \mathbb{R}^K | 1 \le i < j \le B\}$ denotes the trainable categorical logits. We rewrite Eq. (1) and obtain the negative evidence lower bound (ELBO):

$$\mathcal{L}(\theta, w) = -\mathbb{E}_{q(\alpha|\theta)}[\log p(\mathcal{D}|\alpha, w)] + D_{\mathrm{KL}}(q(\alpha|\theta)\|p(\alpha)). \tag{4}$$

Notably, minimizing $\mathcal{L}$ w.r.t. $\theta$ and $w$ corresponds to Bayesian inference on $\alpha$ and maximum a posteriori (MAP) estimation of $w$[2], respectively. Thus, the optimization of the network structure and network weights can be unified as $\min_{\theta, w} \mathcal{L}(\theta, w)$. To resolve this, we relax both $p(\alpha^{(i,j)})$ and $q(\alpha^{(i,j)}|\theta^{(i,j)})$ to be the concrete distributions (Maddison et al., 2016). Then, samples $\alpha$ from $q(\alpha|\theta)$ are generated via the softmax transformation:

$$\alpha = g(\theta, \epsilon) = \{\mathrm{softmax}((\theta^{(i,j)} + \epsilon^{(i,j)})/\tau)\}, \tag{5}$$

where $\epsilon = \{\epsilon^{(i,j)} \in \mathbb{R}^K | \epsilon_k^{(i,j)} \sim \text{Gumbel i.i.d.}\}$ are the Gumbel variables and $\tau \in \mathbb{R}_+$ is the temperature. Then we derive the following gradient estimators:

$$\nabla_\theta \mathcal{L}(\theta, w) = \mathbb{E}_\epsilon[-\nabla_\theta \log p(\mathcal{D}|g(\theta, \epsilon), w) + \nabla_\theta \log q(g(\theta, \epsilon)|\theta) - \nabla_\theta \log p(g(\theta, \epsilon))], \tag{6}$$
$$\nabla_w \mathcal{L}(\theta, w) = \mathbb{E}_\epsilon[-\nabla_w \log p(\mathcal{D}|g(\theta, \epsilon), w)]. \tag{7}$$

The first term in Eq. (6) corresponds to the gradient of the negative log likelihood and we leave how to estimate the last two terms (i.e. log densities) in the next section. In practice, we approximate the expectation in Eq. (6) and Eq. (7) with $T$ Monte Carlo (MC) samples, and update the structure and the weights $w$ simultaneously.

After training, we gain the following predictive distribution:

$$p(y|x_{new}, w^*) = \mathbb{E}_{q(\alpha|\theta^*)}[p(y|x_{new}, \alpha, w^*)], \tag{8}$$

where $\theta^*$ and $w^*$ denote the converged parameters. Eq. (8) implies that the model predicts by ensembling the predictions of the networks whose structures are randomly sampled.

---

[1] We will refer to the cell-based differentiable NAS as NAS for short if there is no misleading.

[2] This is because we use regularizor on weights, e.g., weight decay, to alleviate over-fitting.

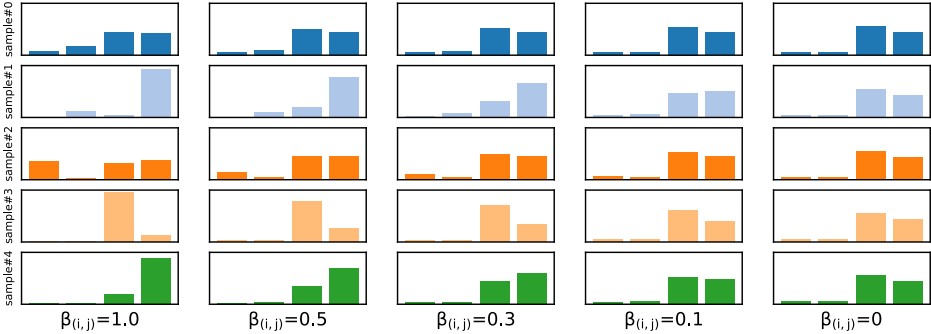

Figure 2: Each column includes 5 samples $\boldsymbol{\alpha}^{(i,j)}$ from an adaptive concrete distribution with some $\boldsymbol{\beta}^{(i,j)}$ at $\tau = 1$. Samples in every row share the same $\boldsymbol{\epsilon}^{(i,j)}$. The base class probabilities are $\mathrm{softmax}(\boldsymbol{\theta}^{(i,j)}) = [0.05, 0.05, 0.5, 0.4]$ in each sample.

### 3.1 ADAPTIVE CONCRETE DISTRIBUTION

The weight sharing mechanism in DBSN is a non-trivial contribution for Bayesian structure learning, enabling computationally efficient optimization. But it also causes unignorable training challenges. Specifically, because of the limited capacity of the shared weights $\boldsymbol{w}$, we have challenges to train it sufficiently well to be suitable for all the structures. The under-fitting of $\boldsymbol{w}$ then brings bias in the learning of $\boldsymbol{\alpha}$'s variational posterior and results in unsatisfying convergence of the whole model. We note that an analogous phenomenon was also observed by Mackay et al. (2019) in the gradient-based hyper-parameter optimization scenario.

Therefore, to facilitate $\boldsymbol{w}$ to fit the structure distribution better and eventually benefit the Bayesian structure learning, we expect to reduce the variance of the structure distribution. Specifically, we analyze the reparameterization procedure of the concrete distribution, and propose to multiply a tunable scalar $\boldsymbol{\beta}^{(i,j)}$ with $\boldsymbol{\epsilon}^{(i,j)}$ in the sampling:

$$\boldsymbol{\alpha}^{(i,j)} = g(\boldsymbol{\theta}^{(i,j)}, \boldsymbol{\beta}^{(i,j)}, \boldsymbol{\epsilon}^{(i,j)}) = \mathrm{softmax}((\boldsymbol{\theta}^{(i,j)} + \boldsymbol{\beta}^{(i,j)}\boldsymbol{\epsilon}^{(i,j)})/\tau). \quad (9)$$

Accordingly, we derive the log probability density of the adaptive concrete distribution which is slightly different from that of the concrete distribution (*see the detailed derivation in Appendix A*):

$$\log p(\boldsymbol{\alpha}^{(i,j)}|\boldsymbol{\theta}^{(i,j)}, \boldsymbol{\beta}^{(i,j)}) = \log((K-1)!) + (K-1)\log\tau - (K-1)\log\boldsymbol{\beta}^{(i,j)}$$
$$- \sum_{k=1}^{K}\log\boldsymbol{\alpha}_k^{(i,j)} + \sum_{k=1}^{K}\left[\frac{\boldsymbol{\theta}_k^{(i,j)} - \tau\log\boldsymbol{\alpha}_k^{(i,j)}}{\boldsymbol{\beta}^{(i,j)}}\right] - K * \mathrm{L\Sigma E}_{k=1}^{K}\left[\frac{\boldsymbol{\theta}_k^{(i,j)} - \tau\log\boldsymbol{\alpha}_k^{(i,j)}}{\boldsymbol{\beta}^{(i,j)}}\right], \quad (10)$$

where L$\Sigma$E represents the log-sum-exp operation. With this, the last two terms of Eq. (6) can be estimated exactly.

Obviously, the adaptive concrete distribution degrades to the concrete distribution when $\boldsymbol{\beta}^{(i,j)} = 1$. As shown in Figure 2, sliding $\boldsymbol{\beta}^{(i,j)}$ from 1 to 0 decreases the diversity of the sampled structures gradually. Therefore, we should also keep $\boldsymbol{\beta}^{(i,j)}$ from being too small to avoid the over-fitting issue which the point-estimate structure (i.e., $\boldsymbol{\beta}^{(i,j)} = 0$) may suffer from. In practice, we choose to gradually reduce the sample variance along with the convergence of the weights, by decaying $\boldsymbol{\beta}^{(i,j)}$ from 1 to 0.5 with a linear schedule in the training.

### 3.2 PRACTICAL IMPROVEMENTS OF THE STRUCTURE LEARNING SPACE

In order to make the training more stable and more efficient, we modify some changes to the structure learning space (i.e., the support of the structure distribution) commonly adopted in NAS.

**Overall modification.** To facilitate more effective information flow in the cell, we let the input of a cell (i.e., the output of the previous cell) be fixedly connected to all the internal nodes by $1 \times 1/3 \times 3$ convolutions in the classification/segmentation tasks. We only learn the connections between the $B$ internal nodes, as shown in Appendix F. The resulted nodes are concatenated along with the input to get the cell's output. In spirit of DenseNet (Huang et al., 2017) and FC-DenseNet (Jégou

et al., 2017), we constrain the downsampling/upsampling modules to be the typical BN-ReLU-Conv-Pooling/ConvTranspose operations, to ease the learning of the network structure.

**Batch normalization.** NAS usually adopts the order of ReLU-Conv-BN in operations. However, in the searching stage, the learnable affine transformations in batch normalizations are always disabled to avoid the output rescaling issue (Liu et al., 2019). NAS does not suffer from this since it trains another network with learnable batch normalizations in the extra re-training stage. Instead, DBSN has to fix the issue because we do not re-train the model. Thus, we propose to put a complete batch normalization in the front of the next layer. Namely, we adopt the BN-ReLU-Conv-BN convolutional layers, where the first BN has learnable affine parameters while the second one does not.

**Candidate operations.** In order to make the training more efficient, we remove the operations which are popular in NAS but unnecessary in DBSN, including all the 5×5 convolutions that can be replaced by stacked 3×3 convolutions, and all the pooling layers which are mainly used for the downsampling module. Then, the candidate operations in DBSN are: 3×3 separable convolutions, 3×3 dilated separable convolutions, identity and *zero*. We follow Liu et al. (2019) for the detailed settings of these operations.

**Group operation.** To obtain the $j^{th}$ node in a cell, there are $(j-1)K$ operations from its predecessors to calculate, which can be organized into $K$ groups according to the operation type. Note that the operations in a group are independent, so we advocate replacing them with a group operation (e.g., group convolution), which improves the efficiency significantly.

### 3.3 DISCUSSION

One may concern that the practical choice of weight sharing could push the structure distribution toward the most likely point for the weights and result in a Dirac structure distribution. However, the prior keeps the variational posterior from collapsing via a KL regularization (last term of Eq. (4)). Besides, recall that $w$ is a set including the parameters of all the redundant operations. Then, in fact, different network structures adjust w.r.t. different subsets of $w$, further alleviating the structure collapsing issue. The widely used technique of MC Dropout (Gal & Ghahramani, 2016; Gal et al., 2017) can also be seen as using the same weights for different structures. Their empirical results also prove that this kind of model choice is reasonable. Nevertheless, capturing the dependency of $w$ on $\alpha$ may indeed bring more accurate modeling and we leave this as future work.

We also emphasize that using point estimates for the weights benefits the whole model's learning significantly. On one hand, as stated in the introduction, there are still frustrating difficulties to achieve scalable Bayesian inference on the high-dimensional network weights, which is also proven by the results in Table 1, Table 3, and Appendix C. On the other hand, DBSN deploys weight decay regularizor on weights, which implicitly imposes a Gaussian prior on $w$. Then, DBSN performs maximum a posteriori (MAP) estimation of $w$, namely, estimating the mode of $w$'s posterior distribution $p(w|\mathcal{D})$, which can be viewed as doing approximate Bayesian inference on $w$.

## 4 RELATED WORK

Learning flexible Bayesian models has long been the goal of the community (MacKay, 1992; Neal, 1995; Balan et al., 2015; Wang & Yeung, 2016). The stochastic variational inference methods for Bayesian neural networks are particularly appealing owing to their analogy to the ordinary back-propagation (Graves, 2011; Blundell et al., 2015). More expressive distributions, such as matrix-variate Gaussians (Sun et al., 2017) or multiplicative normalizing flows (Louizos & Welling, 2017), have also been introduced to represent the posterior dependencies, but they are hard to train without heavy approximations. Recently, there is an increasing interest in developing Adam-like optimizers to perform natural-gradient variational inference for BNNs (Zhang et al., 2018; Bae et al., 2018; Khan et al., 2018). Despite enabling the scalability, these methods seem to demonstrate compromising performance compared to the state-of-the-art deep models. Interpreting the stochastic techniques of the deep models as Bayesian inference is also insightful (Gal & Ghahramani, 2016; Kingma et al., 2015; Teye et al., 2018; Mandt et al., 2017; Lakshminarayanan et al., 2017), but these methods still have relatively restricted and inflexible posterior approximations. Dikov & Bayer (2019) propose a unified Bayesian framework to infer the posterior of both the network weights and the structure, which is most similar to DBSN, but the network structure considered by them, namely layer size

and network depth, is essentially impractical for complicated deep models. Instead, we inherit the design of the structure learning space for NAS, and provide insightful techniques to improve the convergence, thus enabling effective Bayesian structure learning for deep neural networks.

Neural architecture search (NAS) has drawn tremendous attention, where reinforcement learning (Zoph & Le, 2016; Zoph et al., 2018; Pham et al., 2018), evolution (Real et al., 2019) and Bayesian optimization (Kandasamy et al., 2018) have all been introduced to solve it. More recently, differentiable NAS (Liu et al., 2019; Xie et al., 2019; Cai et al., 2019; Wu et al., 2019) is attractive because it reduces the prohibitive computational cost immensely. However, existing differentiable NAS methods search the network structure in a meta-learning way (Finn et al., 2017), and need to re-train another network with the pruned compact structure after the searching. In contrast, DBSN unifies the learning of weights and structure in one training stage, alleviating the mismatch of structures during the search and re-training, as well as inefficiency issues suffered by differentiable NAS.

## 5 EXPERIMENTS

To validate the structure learning ability and the predictive performance of DBSN, we first evaluate it on image classification and segmentation tasks. For the estimation of the predictive uncertainty, we concern model calibration and generalization of the predictive uncertainty to adversarial examples as well as out-of-distribution samples, following existing work. We show that DBSN outperforms strong baselines in these tasks, shedding light on practical Bayesian deep learning.

### 5.1 IMAGE CLASSIFICATION ON CIFAR-10 AND CIFAR-100

**Setup.** We set $B = 7$, $T = 4$ and $K = 4$, thus, $\alpha$ consists of $7 \times 6/2 = 21$ sub-variables. The whole network is composed of 12 cells and 2 downsampling modules which have a channel compression factor of 0.4 and are located at the 1/3 and 2/3 depth. We employ a $3 \times 3$ convolution before the first cell and put a global average pooling followed by a fully connected (FC) layer after the last cell. The redundant operations all have 16 output channels. We initialize $w$ and $\theta$ following He et al. (2015) and Liu et al. (2019), respectively. The prior distributions of $\alpha^{(i,j)}$ are set to be concrete distributions with uniform class probabilities. A momentum SGD with initial learning rate 0.1 (divided by 10 at 50% and 75% of the training procedure following (Huang et al., 2017)), momentum 0.9 and weight decay $10^{-4}$ is used to train the weights $w$. An Adam optimizer with learning rate $3 \times 10^{-4}$, momentum (0.5, 0.999) is used to learn $\theta$. We deploy the standard data augmentation scheme (mirroring/shifting) and normalize the data with the channel statistics. The whole training set is used for optimization. We train DBSN for 100 epochs with batch size 64, which takes one day on 4 GTX 1080-Tis. The implementation depends on PyTorch (Paszke et al., 2017) and the codes are available online at `https://github.com/anonymousest/DBSN`.

**Baselines.** Besides comparison to the advanced deep models, we also design a series of baselines for fair comparisons. 1) **DBSN\***: we substitute the concrete distribution for the adaptive concrete distribution. 2) **DBSN-1**: we use $T = 1$ sample in the gradient estimation. 3) **Fixed $\alpha$**: we fix the structure of the network by setting the weight of every operation to be $1/K$. 4) **Dropout**: based on Fixed $\alpha$, we further add dropout on every computational node with a drop rate of 0.2. 5) **Drop-path**: based on Fixed $\alpha$, we further apply drop-path (Larsson et al., 2016) regularisation on the convolutional redundant operations with a path drop rate of 0.3. 6) **Random $\alpha$**: we fix the distributions of $\alpha^{(i,j)}$ as concrete distributions with uniform class probabilities and only train $w$ with randomly sampled $\alpha$. 7) **PE**: we view the structure as point estimates and train it as well as $w$ simultaneously. 8) **DARTS**: we view the structure as point estimates but we train it on half of the training set while train $w$ on the other half, resembling the first order DARTS (Liu et al., 2019). 9) **NEK-FAC**: we train a VGG16 network with weight uncertainty using the noisy EK-FAC (Bae et al., 2018) and the corresponding default settings. 10) **BNN-LS**: we replace all the convolutional and fully connected layers in PE with their Bayesian counterparts to build a BNN with Learnable Structure. 11) **Fully Bayesian DBSN**: we replace all the convolutional and fully connected layers in DBSN with their Bayesian counterparts to build a Fully Bayesian neural network. In BNN-LS and Fully Bayesian DBSN, we employ fully factorized Gaussian distributions on weights and adopt BBB (Blundell et al., 2015) for inference. When testing DBSN, DBSN\*, DBSN-1, Random $\alpha$, NEK-FAC, Dropout, Drop-path, BNN-LS and Fully Bayesian DBSN, we ensemble the predictive

Table 1: Comparison with competing baselines in terms of the number of parameters and test error rate. DBSN and its variants have 1.1 M parameters on CIFAR-100 due to a larger FC layer.

| Method | Params (M) | CIFAR-10 (%) | CIFAR-100 (%) |
|---|---|---|---|
| **ResNet (He et al., 2016a)** | 1.7 | 6.61 | - |
| **Stochastic Depth (Huang et al., 2016)** | 1.7 | 5.23 | 24.58 |
| **ResNet (pre-activation) (He et al., 2016b)** | 1.7 | 5.46 | 24.33 |
| **DenseNet (Huang et al., 2017)** | 1.0 | 5.24 | 24.42 |
| **DenseNet-BC (Huang et al., 2017)** | 0.8 | **4.51** | **22.27** |
| **NEK-FAC (Bae et al., 2018)** | 3.7 | 7.43 | 37.47 |
| **BNN-LS** | 2.0 | $9.85 \pm 0.42$ | $30.98 \pm 0.36$ |
| **Fully Bayesian DBSN** | 2.0 | $9.57 \pm 0.55$ | $31.39 \pm 0.06$ |
| **DBSN** | 1.0 | $\mathbf{4.98} \pm 0.24$ | $\mathbf{22.50} \pm 0.26$ |
| **DBSN\*** | 1.0 | $5.22 \pm 0.34$ | $22.78 \pm 0.19$ |
| **DBSN-1** | 1.0 | $5.60 \pm 0.17$ | $23.44 \pm 0.28$ |
| **Fixed $\alpha$** | 1.0 | $5.66 \pm 0.24$ | $24.27 \pm 0.15$ |
| **Random $\alpha$** | 1.0 | $6.12 \pm 0.12$ | $23.60 \pm 0.19$ |
| **Dropout** | 1.0 | $5.83 \pm 0.19$ | $23.67 \pm 0.28$ |
| **Drop-path** | 1.0 | $5.77 \pm 0.05$ | $23.12 \pm 0.13$ |
| **PE** | 1.0 | $5.79 \pm 0.34$ | $24.19 \pm 0.17$ |
| **DARTS** | 1.0 | $8.91 \pm 0.16$ | $31.87 \pm 0.12$ |

probabilities from 100 random runs (we adopt this strategy in all the following experiments, unless stated otherwise).

We repeat every experiment 3 times and report the averaged error rate and standard deviation in Table 1. Notably, DBSN demonstrates comparable performance with state-of-the-art deep neural networks. DBSN outperforms the powerful ResNet (He et al., 2016a) and DenseNet (Huang et al., 2017) with statistical evidence, and only presents modestly higher error rates than those of DenseNet-BC (Huang et al., 2017), which probably results from the usage of the expressive and efficient bottleneck layer in DenseNet-BC. This comparison highlights the practical value of DBSN.

Comparisons between DBSN and the baselines designed by ourselves are more insightful and convincing. 1) DBSN surpasses DBSN\*, revealing the effectiveness of the adaptive concrete distribution. 2) DBSN-1 is remarkably worse than DBSN owing to the higher variance of the estimated gradients with only one sample. 3) Comparison of DBSN and Fixed $\alpha$ validates that adapting the network structure w.r.t. the data distribution benefits the fitting of the model, resulting in substantially enhanced performance. 4) Random $\alpha$, Dropout, and Drop-path train the networks with manually-designed untunable randomness, and hence are inferior to DBSN. 5) NEK-FAC gains rather compromising performance, with the powerful VGG16 architecture and one of the most advanced variational BNNs algorithms, suggesting us to prefer DBSN instead of the classic BNNs in the scenarios where the performance is a major concern. 6) BNN-LS and Fully Bayesian DBSN both have poor performance, due to the fundamental difficulties of modeling distributions over high dimensional weights. 7) PE and DARTS are two methods to learn the point-estimate network structure, both of which fall behind in terms of the test error. In particular, DARTS is much worse as it only trains the weights on half of the training set. This shows that DBSN is an appealing choice for effective neural structure learning with only one-stage training.

## 5.2 SEMANTIC SEGMENTATION ON CAMVID

To further verify that learning the network structure w.r.t. the data helps DBSN to obtain better performance than the standard NNs and BNNs, we apply DBSN to the challenging segmentation benchmark CamVid (Brostow et al., 2008). Our implementation is based on the brief FC-DenseNet framework (Jégou et al., 2017). Specifically, we only replace the original dense blocks with the structure-learnable cells, without introducing further advanced techniques from the semantic segmentation community, to figure out the performance gain only resulted from the learnable network structure. For the setup, we set $B = 5$ (same as the number of layers in every dense block of FC-DenseNet67) and $T = 1$, and learn two cell structures for the downsampling path and upsampling

Table 2: Comparison of semantic segmentation performance on CamVid dataset. * indicates results from our implementation.

| Method | Pretrained | Params (M) | Mean IoU | Global accuracy |
|---|---|---|---|---|
| **SegNet** (Badrinarayanan et al., 2015) | ✓ | 29.5 | 46.4 | 62.5 |
| **Bayesian SegNet** (Kendall et al., 2015) | ✓ | 29.5 | 63.1 | 86.9 |
| **FC-DenseNet67** (Jégou et al., 2017) | ✗ | 3.5 | 63.1* | 90.4* |
| **DBSN** | ✗ | 3.3 | **65.4** | **91.4** |

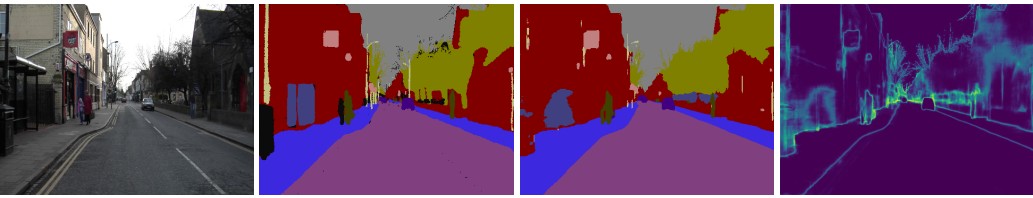

Figure 3: Visualization of the segmentation and uncertainty results of DBSN on CamVid. From left to right: original image, ground-truth segmentation, the estimated segmentation, and pixel-wise predictive uncertainty. The black color in ground-truth labels represents the background (void) class.

path, respectively. We use a momentum SGD with initial learning rate 0.01 (which decays linearly after 350 epochs), momentum 0.9 and weight decay $10^{-4}$ instead of the original RMSprop for better results. The other settings follow Jégou et al. (2017) and the classification experiments above. We also implement FC-DenseNet67 as a baseline. We present the results in Table 2 and Figure 3.

It is evident that DBSN surpasses the competing FC-DenseNet67 by a large margin while using fewer parameters. DBSN also demonstrates significantly better performance than the classic Bayesian SegNet which adopts MC dropout for uncertainty estimation. We emphasize this experiment shows that the proposed approach is generally applicable. It is also worth noting that the uncertainty produced by DBSN is interpretable (see Figure 3): the edges of the objects and the regions which contain overlapping have substantially higher uncertainty than the other parts.

### 5.3 ESTIMATION OF PREDICTIVE UNCERTAINTY

To validate that DBSN can provide promising predictive uncertainty, we evaluate it via calibration. We further examine the predictive uncertainty on adversarial examples and out-of-distribution (OOD) samples to test if the model *knows what it knows*. We also pay particular attention to the comparison between Drop-path and Dropout to double-check if more structured randomness (Larsson et al., 2016) benefits predictive uncertainty more.

Calibration is orthogonal to the accuracy (Lakshminarayanan et al., 2017) and can be well estimated by the Expected Calibration Error (ECE) (Guo et al., 2017). Thus, we evaluate the trained models on the test set of CIFAR-10 and CIFAR-100 and calculate their ECE, as shown in Table 3. We also plot some reliability diagrams (Guo et al., 2017) in Appendix D, to provide a direct explanation of calibration. Unsurprisingly, DBSN achieves state-of-the-art calibration. DBSN outperforms the strong baselines, Dropout and NEK-FAC. NEK-FAC, BNN-LS and Fully Bayesian DBSN all have much worse ECE than DBSN, implying structure uncertainty's superiority over weight uncertainty. We also notice that Drop-path is better than Dropout in terms of ECE, supporting our hypothesis that more structured randomness is more beneficial to the predictive uncertainty.

To test the predictive uncertainty on the adversarial examples, we apply the fast gradient sign method (FGSM) (Goodfellow et al., 2014) to attack the trained models on CIFAR-10 and CIFAR-100 using the corresponding test samples[3]. Then we calculate the predictive entropy of the generated adversarial examples and depict the average entropy in Figure 4. As expected, the entropy of DBSN grows rapidly as the perturbation size increases, implying DBSN becomes pretty uncertain when encoun-

---

[3]For DBSN, DBSN*, Random $\alpha$, NEK-FAC, Dropout, and Drop-path, we attack using the ensemble of predictions from 30 stochastic runs and then we test the manipulated adversarial examples with 30 runs as well.

Table 3: Comparison of model calibration in terms of the Expected Calibration Error (ECE). Smaller is better.

| Dataset | DBSN | DBSN* | Fixed $\alpha$ | Dropout | Drop-path | NEK-FAC | BNN-LS | Fully Bayesian DBSN |
|---|---|---|---|---|---|---|---|---|
| CIFAR-10 | **0.0109** | 0.0111 | 0.0327 | 0.0150 | 0.0133 | 0.0434 | 0.0745 | 0.0966 |
| CIFAR-100 | **0.0599** | 0.0677 | 0.1259 | 0.0617 | **0.0524** | 0.1665 | 0.0700 | 0.1091 |

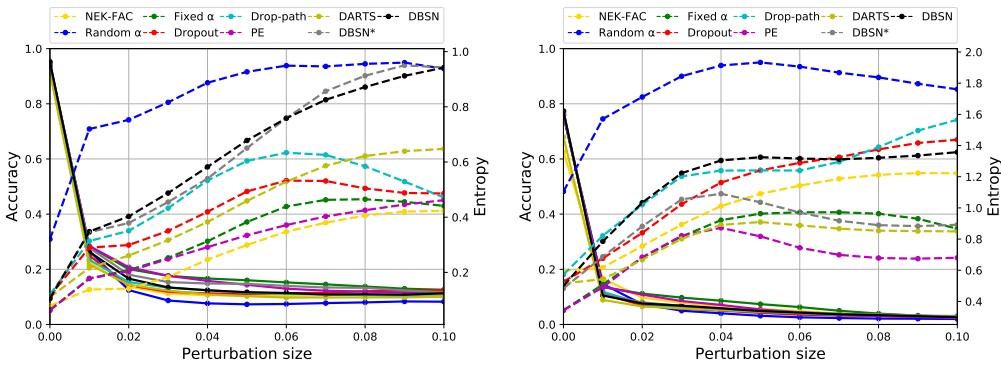

Figure 4: Accuracy (solid) and entropy (dashed) vary w.r.t. the adversarial perturbation size on CIFAR-10 (left) and CIFAR-100 (right).

tering adversarial perturbations. By contrast, the change in entropy of Dropout and NEK-FAC is relatively moderate, which means that these methods are not as sensitive as DBSN to the adversarial examples. Besides, Drop-path is still better than Dropout, consistent with the conclusion above. We also note that Random $\alpha$ has the highest predictive entropy. We speculate that this is because Random $\alpha$ adopts the most diverse network structures (which results from the uniform class probabilities), and the ensemble of predictions from the corresponding networks is easier to be uniform. We further attack with more powerful algorithms, e.g., the Basic Iterative Method (BIM) (Kurakin et al., 2016), and provide the results in Appendix E.

Moreover, we look into the entropy of the predictive distributions on OOD samples, to adequately evaluate the quality of uncertainty estimation. We use the trained models on CIFAR-10 and CIFAR-100, and take the samples from the test set of SVHN as OOD samples. We calculate their predictive entropy and draw the empirical CDF of the entropy in Figure 5, following Louizos & Welling (2017). The curve close to the bottom right corner is expected as it means most OOD samples have relatively large entropy (i.e., low prediction confidence). Obviously, DBSN demonstrates comparable or even better results than the competing methods like Dropout and NEK-FAC. In addition, Drop-path attains substantially improved results than Dropout. Analogous to the experiments on adversarial examples, Random $\alpha$ provides impressive predictive uncertainty on the OOD samples.

In conclusion, DBSN consistently delivers state-of-the-art predictive uncertainty in various scenarios, validating the effectiveness of structure uncertainty.

## 5.4 RETHINKING OF THE ONE-SHOT NAS

One-shot NAS (Bender et al., 2018; Guo et al., 2019) first trains the weights of a super network and then searches for a good structure given the weights. This avoids the bias induced by the gradient-based joint optimization of the differentiable NAS. However, we argue that the super network trained with the fixed (Bender et al., 2018) or uniformly sampled (Guo et al., 2019) network structures cannot flexibly focus its capacity on the most crucial operations, harming the subsequent searching. To this end, we have conducted a set of experiments to check whether dynamically adjusting the network structure at the stage of weight training helps to find better network structures eventually. Observing that DBSN trains a super network with adaptive network structures and Random $\alpha$ trains a super network with unadjustable structures (similar to the uniform sampling used by Guo et al. (2019)), we choose to search for the optimal structure distributions based on the trained weights

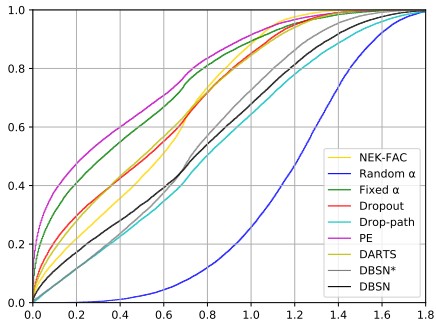 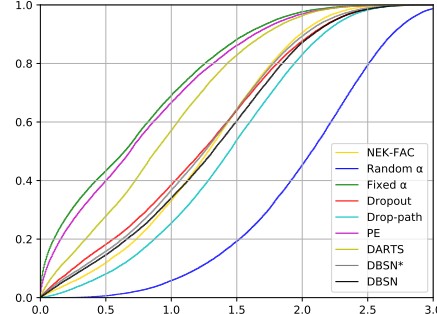

Figure 5: Empirical CDF for the entropy of the predictive distributions on SVHN dataset of models trained on CIFAR-10 (left) and CIFAR-100 (right). The curves that are closer to the bottom right corner are better.

Table 4: Comparison of the searched structure distributions based on the trained network weights from DBSN and Random $\alpha$.

|  | DBSN | Random $\alpha$ |
|---|---|---|
| Test error (%) | **5.46** | 5.98 |

from DBSN and Random $\alpha$[4]. After searching, we train new networks with the searched structure distributions (fixed in the training) from scratch, and then test their performance. The results are shown in Table 4. The searched structure distribution based on the weights trained by DBSN outperforms the other one significantly, supporting our hypotheses. Therefore, we propose to reasonably adapt the structure in the weight-training stage of one-shot NAS, which drives the most useful operations to be optimized thoroughly and eventually yields more powerful network structures.

## 5.5 VISUALIZATION OF THE LEARNED STRUCTURE DISTRIBUTIONS

We visualize the learned structure distributions in Appendix F. The structure distributions for different tasks look quite different, which implies that the structures are learned in a way that accounts for the specific characteristics in the data.

## 6 CONCLUSION

In this work, we have introduced a novel Bayesian structure learning approach for deep neural networks. The proposed DBSN draws the inspiration from the network design of NAS and models the network structure as Bayesian variables. Stochastic variational inference is employed to jointly learn the network weights and the distribution of the network structure. We further develop the adaptive concrete distribution and improve the structure learning space to facilitate the convergence of the whole model. Empirically, DBSN has revealed impressive performance on the discriminative learning tasks, surpassing the advanced deep models, and presented state-of-the-art predictive uncertainty in various scenarios. In conclusion, DBSN provides a more practical way for Bayesian deep learning, without compromise between the predictive performance and the Bayesian uncertainty.

There are two major directions for future work. On one hand, the current DBSN is not efficient enough, so some strategies need to be discovered to make DBSN more efficient. On the other hand, DBSN still has a relatively restricted structure learning space. Thus, more operations can be introduced and more global network structures can be learned in future work.

---

[4]We initialize $\theta^{(i,j)}$ randomly and initialize $\beta^{(i,j)}$ with 1. Given the fixed network weights, we optimize $\theta^{(i,j)}$ and $\beta^{(i,j)}$ by gradient descent. The searching lasts for 20 epochs.

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

## A    DERIVATION OF THE LOG PROBABILITY DENSITY OF THE ADAPTIVE CONCRETE DISTRIBUTION

For clear expression, We simply denote $\boldsymbol{\alpha}^{(i,j)}$, $\boldsymbol{\theta}^{(i,j)}$, $\boldsymbol{\beta}^{(i,j)}$ and $\boldsymbol{\epsilon}^{(i,j)}$ as $\boldsymbol{\alpha}$, $\boldsymbol{\theta}$, $\beta$ and $\boldsymbol{\epsilon}$, respectively. Let $\boldsymbol{p} = \mathrm{softmax}(\boldsymbol{\theta})$. Consider

$$\boldsymbol{\alpha}_k = \frac{\exp((\boldsymbol{\theta}_k + \beta\boldsymbol{\epsilon}_k)/\tau)}{\sum_{i=1}^{K}\exp((\boldsymbol{\theta}^i + \beta\boldsymbol{\epsilon}^i)/\tau)} = \frac{\exp((\log\boldsymbol{p}_k + \beta\boldsymbol{\epsilon}_k)/\tau)}{\sum_{i=1}^{K}\exp((\log\boldsymbol{p}^i + \beta\boldsymbol{\epsilon}^i)/\tau)}.$$

Let $\boldsymbol{z}_k = \log\boldsymbol{p}_k + \beta\boldsymbol{\epsilon}_k = \log\boldsymbol{p}_k - \beta\log(-\log(\boldsymbol{u}_k))$, where $\boldsymbol{u}_k \sim \mathcal{U}(0,1)$ i.i.d.. It has density

$$\frac{1}{\beta}\boldsymbol{p}_k^{1/\beta}\exp(-\frac{\boldsymbol{z}_k}{\beta})\exp(-\boldsymbol{p}_k^{1/\beta}\exp(-\frac{\boldsymbol{z}_k}{\beta})).$$

We denote $c = \sum_{i=1}^{K}\exp(\boldsymbol{z}_i/\tau)$, then $\boldsymbol{\alpha}_k = \exp(\boldsymbol{z}_k/\tau)/c$. We consider this transformation:

$$F(\boldsymbol{z}_1,\ldots,\boldsymbol{z}_K) = (\boldsymbol{\alpha}_1,\ldots,\boldsymbol{\alpha}_{K-1},c),$$

which has the following inverse transformation:

$$F^{-1}(\boldsymbol{\alpha}_1,\ldots,\boldsymbol{\alpha}_{K-1},c) = (\tau(\log\boldsymbol{\alpha}_1 + \log c),\ldots,\tau(\log\boldsymbol{\alpha}_K + \log c)),$$

whose Jacobian has the determinant (refer to the derivation of the concrete distribution (Maddison et al., 2016)):

$$\frac{\tau^K}{c\prod_{i=1}^{K}\boldsymbol{\alpha}_i}.$$

Multiply this with the density of $\boldsymbol{z}$, we get the density

$$\frac{\tau^K\prod_{i=1}^{K}\frac{1}{\beta}\boldsymbol{p}_i^{1/\beta}\exp(-\frac{\tau(\log\boldsymbol{\alpha}_i + \log c)}{\beta})\exp(-\boldsymbol{p}_i^{1/\beta}\exp(-\frac{\tau(\log\boldsymbol{\alpha}_i + \log c)}{\beta}))}{c\prod_{i=1}^{K}\boldsymbol{\alpha}_i}.$$

Let $r = \log c$, then apply the change of variables formula, we obtain the density:

$$\frac{\tau^K\prod_{i=1}^{K}\boldsymbol{p}_i^{1/\beta}}{\beta^K\prod_{i=1}^{K}\boldsymbol{\alpha}_i^{(1+\tau/\beta)}}\exp(-\frac{K\tau r}{\beta})\exp(-\sum_{i=1}^{K}(\boldsymbol{p}_i\boldsymbol{\alpha}_i^{-\tau})^{1/\beta}\exp(-\frac{\tau r}{\beta})).$$

We use $\gamma$ to substitute $\log\sum_{i=1}^{K}(\boldsymbol{p}_i\boldsymbol{\alpha}_i^{-\tau})^{1/\beta}$, then get:

$$\frac{\tau^K\prod_{i=1}^{K}\boldsymbol{p}_i^{1/\beta}}{\exp(\gamma)\beta^K\prod_{i=1}^{K}\boldsymbol{\alpha}_i^{(1+\tau/\beta)}}\exp(\gamma - \frac{K\tau r}{\beta})\exp(-\exp(\gamma - \frac{\tau r}{\beta})).$$

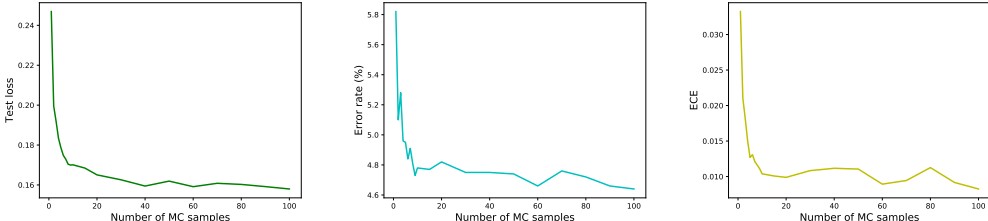

Figure 6: Test loss (left), test error rate (middle), and test ECE (right) of DBSN vary w.r.t. the number of MC samples used in estimating Eq. (8). (CIFAR-10)

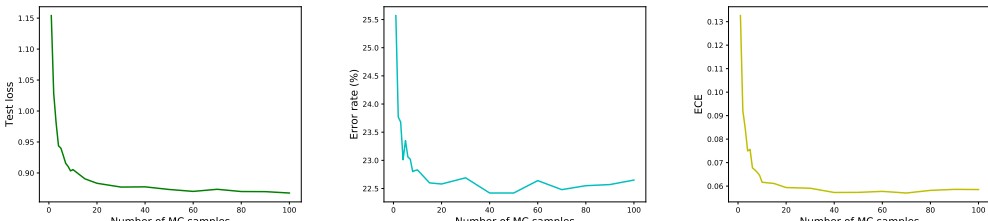

Figure 7: Test loss (left), test error rate (middle), and test ECE (right) of DBSN vary w.r.t. the number of MC samples used in estimating Eq. (8). (CIFAR-100)

Naturally, we can integrate out $r$, and get:

$$
\frac{\tau^K \prod_{i=1}^K \boldsymbol{p}_i^{1/\beta}}{\exp(\gamma)\beta^K \prod_{i=1}^K \boldsymbol{\alpha}_i^{(1+\tau/\beta)}} \left[ \frac{\beta}{\tau} \exp(\gamma - K\gamma)\Gamma(K) \right]
$$
$$
= \frac{\tau^{K-1} \prod_{i=1}^K \boldsymbol{p}_i^{1/\beta}}{\beta^{K-1} \prod_{i=1}^K \boldsymbol{\alpha}_i^{(1+\tau/\beta)}} \exp(-K\gamma)\Gamma(K)
$$
$$
= \frac{((K-1)!)\tau^{K-1}}{\beta^{K-1} \prod_{i=1}^K \boldsymbol{\alpha}_i} \times \frac{\prod_{i=1}^K (\boldsymbol{p}_i \boldsymbol{\alpha}_i^{-\tau})^{1/\beta}}{(\sum_{i=1}^K (\boldsymbol{p}_i \boldsymbol{\alpha}_i^{-\tau})^{1/\beta})^K}.
$$

Then, the log density is:

$$
\log((K-1)!) + (K-1)\log\frac{\tau}{\beta} - \sum_{i=1}^K \log\boldsymbol{\alpha}_i + \sum_{i=1}^K \frac{\log\boldsymbol{p}_i - \tau\log\boldsymbol{\alpha}_i}{\beta} - K * \mathop{\mathrm{L\Sigma E}}_{i=1}^K \frac{\log\boldsymbol{p}_i - \tau\log\boldsymbol{\alpha}_i}{\beta}
$$
$$
= \log((K-1)!) + (K-1)\log\frac{\tau}{\beta} - \sum_{i=1}^K \log\boldsymbol{\alpha}_i + \sum_{i=1}^K \frac{\boldsymbol{\theta}_i - \tau\log\boldsymbol{\alpha}_i}{\beta} - K * \mathop{\mathrm{L\Sigma E}}_{i=1}^K \frac{\boldsymbol{\theta}_i - \tau\log\boldsymbol{\alpha}_i}{\beta},
$$

which is equal to Eq. (10).

## B  THE EFFECTS OF THE NUMBER OF MC SAMPLES IN TEST PHASE

We draw the change of test loss, test error rate and test ECE with respect to the number of MC samples used for testing DBSN in Figure 6 (CIFAR-10) and Figure 7 (CIFAR-100). It is clear that ensembling the predictions from models with various sampled network structures enhances the final predictive performance and calibration significantly. This is in marked contrast to the situation of classic variational BNNs, where using more MC samples does not necessarily bring improvement over using the most likely sample. As shown in the plots, we would better utilize 20+ MC samples to predict the unseen data, for adequately exploiting the learned structure distribution. Indeed, we use 100 MC samples in all the experiments, except the adversarial attack experiments where we use 30 MC samples for attacking and evaluation.

Table 5: Comparison with competing baselines which deploy uncertainty on weights and adopt Adam-like VOGN (Khan et al., 2018) method for inference. (CIFAR-10)

| | Training time (hours) | Test error rate (%) | ECE |
|---|---|---|---|
| **DBSN** | 1.2 | 9.90 | 0.0070 |
| **BNN-LS (with VOGN)** | 13.0 | 28.4 | 0.5391 |
| **Fully Bayesian DBSN (with VOGN)** | 13.0 | 30.5 | 0.5169 |

## C  MORE COMPARISONS BETWEEN DBSN AND COMPETING BASELINES WITH WEIGHT UNCERTAINTY

We realized the BBB method used for modeling weight uncertainty in BNN-LS and Fully Bayesian DBSN may be restrictive, resulting in such weakness. Therefore, we further implemented these two baselines with a most-recently proposed mean-field natural-gradient variational inference method, called Variational Online Gauss-Newton (VOGN) (Khan et al., 2018; Osawa et al., 2019). VOGN is known to work well with advanced techniques, e.g., momentum, batch normalisation, data augmentation. As claimed by Osawa et al. (2019), VOGN demonstrates comparable results to Adam. Then, we replaced the used BBB (Blundell et al., 2015) in BNN-LS and Fully Bayesian DBSN with VOGN, based on VOGN's official repository (`https://github.com/team-approx-bayes/dl-with-bayes`). With the original network size ($B = 7$, 12 cells), the baselines trained with VOGN needed more than one hour for one epoch. Thus we adopted smaller networks ($B = 4$, 3 cells), which have almost 41K parameters, for the two baselines. We also trained a DBSN in the same setting. The detailed parameters to initialize VOGN are here (`https://github.com/anonymousest/DBSN/blob/master/dbsn/train_bnn_torchsso.py#L220`). The experiments were conducted on CIFAR-10 and the results are provided in Table 5. The predictive performance and uncertainty gaps between DBSN and the two baselines are very huge, which possibly results from the under-fitting of the high-dim weight distributions in BNN-LS and Fully Bayesian DBSN. We believe that our implementation is correct because our results are consistent with the original results in Table 1 of Osawa et al. (2019) (VOGN has 75.48% and 84.27% validation accuracy even with even larger 2.5M AlexNet and 11.1M ResNet-18 architectures). Further, DBSN is much more efficient than them. These comparisons strongly reveal the benefits of modeling structure uncertainty over modeling weight uncertainty, highlighting the practical value of DBSN.

## D  MORE RESULTS FOR CALIBRATION

We plot the reliability diagrams of 4 typical methods, which represent the deep BNN with structure uncertainty, the classic BNN with weight uncertainty, the deterministic NN with MC dropout and the standard NN, respectively, in Figure 8. Obviously, DBSN has better reliability diagrams than NEK-FAC and Dropout, proving the effectiveness of the uncertainty on network structure.

## E  ATTACK WITH BIM

We perform an adversarial attack using BIM algorithm. Concretely, we set the number of iteration to be 3 and set the perturbation size in every step to be 1/3 of the whole perturbation size. The experiments mainly focus on the models trained on CIFAR-10. Figure 9 shows the results. Random $\alpha$, DBSN* and DBSN have increasing entropy when the perturbation size changes from 0 to 0.01, but all the other approaches are attacked successfully with entropy dropping. However, strictly, only the Random $\alpha$ at perturbation size 0.01 provides useful predictive uncertainty, and we can use the entropy to reject the predictions. Therefore, we have to agree that BIM is powerful enough to break all the methods, including DBSN. So we advise adjusting DBSN accordingly (e.g., employing adversarial training, using more robust loss) if we want to use DBSN to defend the adversarial attacks.

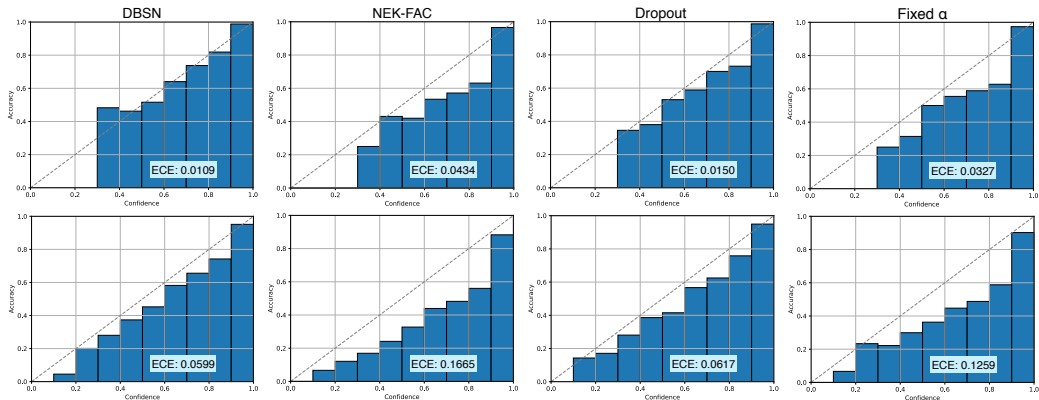

Figure 8: Reliability diagrams for DBSN, NEK-FAC, Dropout and Fixed $\alpha$ on CIFAR-10 (top row) and CIFAR-100 (bottom row). The bars aligning more closely to the diagonal are preferred. Smaller ECE is better.

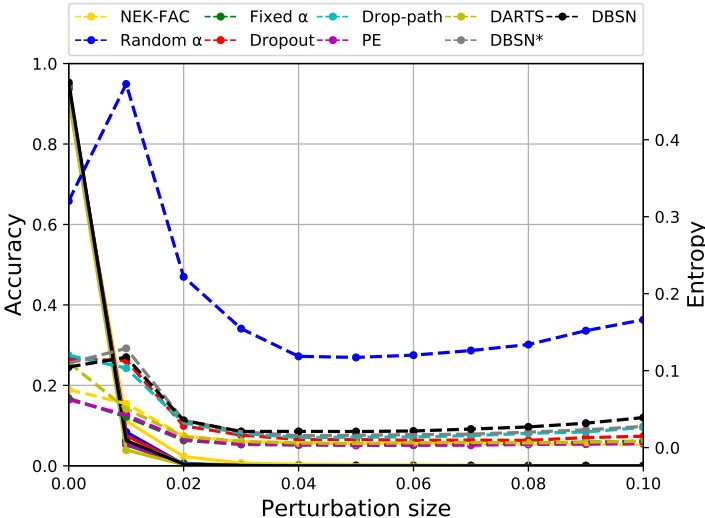

Figure 9: Accuracy (solid) vs entropy (dashed) as a function of the adversarial perturbation size on CIFAR-10. Attack with BIM.

## F    VISUALIZATION OF THE LEARNED STRUCTURES

We visualize the learned structure distributions on different tasks in Figure 10, Figure 11, Figure 12 and Figure 13 (we do not draw the *zero* operation). The structure distributions learned on different tasks look different, validating that DBSN can adapt the network structure according to the data distribution flexibly.

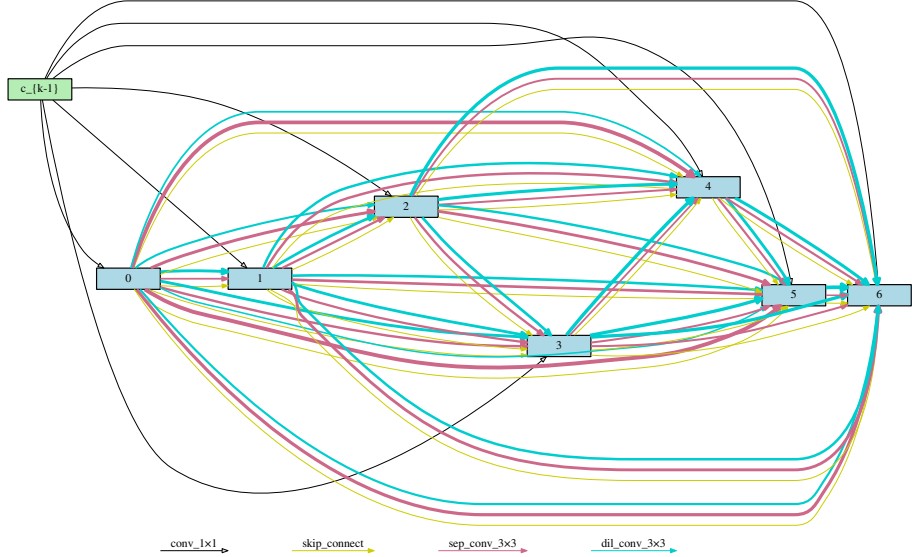

Figure 10: Structure of the cell learned on CIFAR-10. The pen width of an edge implies the sampling probability of its corresponding operation.

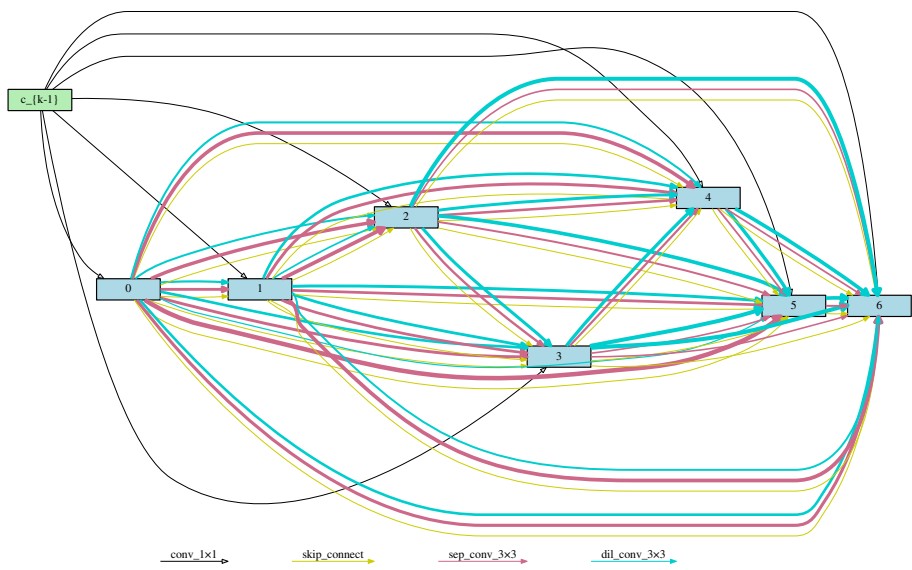

Figure 11: Structure of the cell learned on CIFAR-100. The pen width of an edge implies the sampling probability of its corresponding operation.

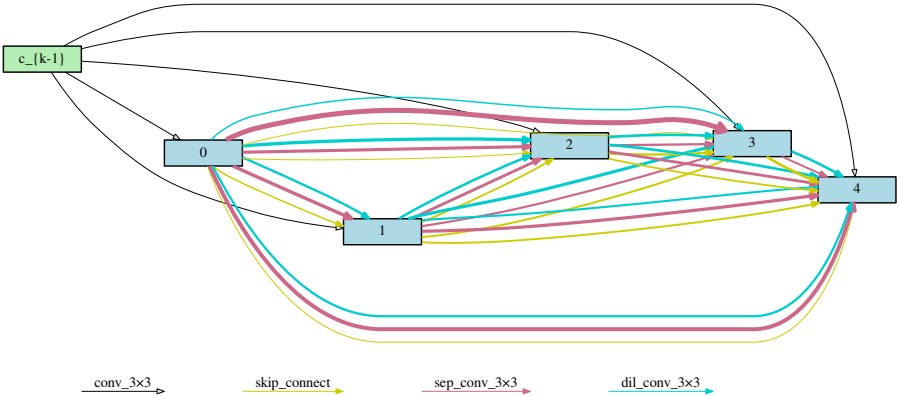

Figure 12: Structure of the cell learned on CamVid (in the downsampling path). The pen width of an edge implies the sampling probability of its corresponding operation.

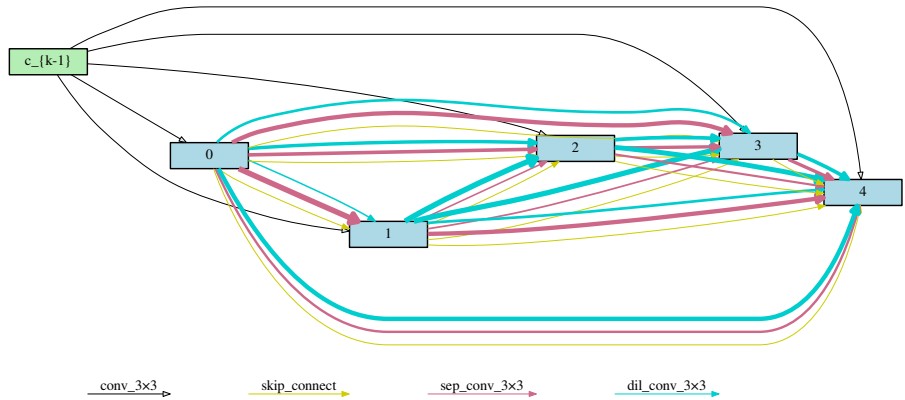

Figure 13: Structure of the cell learned on CamVid (in the upsampling path). The pen width of an edge implies the sampling probability of its corresponding operation.

