# OpenReview forum: "Deep Bayesian Structure Networks"
_ICLR.cc/2020/Conference — Reject_

### Official Review · AnonReviewer2 · 2019-10-22
**Official Blind Review #2**

**Rating:** 3

**Review:**

This paper proposed deep Bayesian structure networks (DBSN) to model weights, \alpha, of the redundant operations in cell-based differentiable NAS. The authors claim that DBSN can achieve better performance (accuracy) than the state of the art.

One of my concerns is the Bayesian formulation introduced in Eq. (4) seems problematic. It is not clear what priors are placed on alpha. In the case of Bayes by BP (BBB), which is cited as Blundell et al. 2015 in the paper, a Gaussian prior (with zero mean) is used. Therefore there is a KL term between the variational distribution q(w) and the prior distribution p(w) to regularize q(w). In DBSN, q(\alpha) is parameterized by \theta and \epsilon, and so is p(\alpha), meaning that the KL term is effectively zero. This is very different from what is done in BBB.

The second major concern is on the experiments. (1) The authors use DARTS as a main baseline and show that DBSN significantly outperforms DARTS. However, looking at the DARTS paper, the test error on CIFAR-10 is around 3% for both the first-order and second-order versions. The test error in Table 1 is around 9%, which is a lot lower. I notice that the DARTS paper has a parameter number of 3.3M, while in the current paper it set to 1M. Given that DARTS is the main baseline method and the same dataset (CIFAR-10) is used, it would make much more sense to use exactly the same architecture for comparison. The current results is hardly convincing. (2) Besides, note that in the DARTS paper, DenseNet-BC has test error of 3.46%, much higher than DARTS (~3%). In Table 2 of this paper however, DARTS is significantly worse than DenseNet-BC (8.91% versus 4.51%). These results are highly inconsistent with previous work.

As mentioned in the paper, Dikov & Bayer 2019 has a very similar idea to perform NAS from a Bayesian perspective. It would be best (and would definitely make the paper stronger) to include some comparison. Even if Dikov & Bayer 2019 is not very scalable, it is at least possible to compare them in smaller network size. Otherwise it is hard to evaluate the contribution of DBSN given this highly similar work.

The authors mentioned in the introduction that DBSN ‘yields more diverse prediction’ and therefore brings more calibrated uncertainty comparing to ensembling different architectures. This is not verified in the experiment section. Table 3 only reports the ECE for one instance of trained networks. For example, it would be interesting to sample different architecture from the alpha learned in DARTS and DBSN, train several networks, ensemble them, and use the variance of the ensemble to compute ECE. This would verify the claim mentioned above.

Do you retrain the network from scratch after the architecture search (which is done in DARTS) for DARTS and DBSN?

I am not convinced by the claim that BNN usually achieve compromising performance. Essentially, BNN, if trained well, is a generalization of deterministic NN. If very flat priors and highly confident variational distributions are used, BNN essentially reduces to deterministic NN.

Missing references on Bayesian deep learning and BNN:

Bayesian Dark Knowledge
Towards Bayesian Deep Learning: A Survey
Natural-Parameter Networks: A Class of Probabilistic Neural Networks

**Experience Assessment:**

I have published in this field for several years.

**Review Assessment: Checking Correctness Of Derivations And Theory:**

I assessed the sensibility of the derivations and theory.

**Review Assessment: Checking Correctness Of Experiments:**

I carefully checked the experiments.

**Review Assessment: Thoroughness In Paper Reading:**

I read the paper thoroughly.

---

> ### Author Response · Authors · 2019-11-13
> **Thank you for the feedback! (Part 2)**
>
> Q5: About “re-train the network from scratch after the architecture search”:
> A: This is not the case for DBSN. As stated above in our response to Q2 and the last parts of Section 2.2&3.3, unlike existing meta-learning approaches in NAS, DBSN does not need to re-train the network after the architecture search. We perform training for once and then can draw multiple samples (i.e., networks with different structures but shared weights) for making predictions. This is exactly one advantage of DBSN over existing NAS solutions.
>
>
> Q6: The claim that BNN usually achieves compromising performance:
> A: Theoretically, considering the over-parameterization nature of modern networks, the data we want to model is always relatively “small” (See e.g. [4] for an informational theoretical analysis of the amount of ‘information’ in a dataset). Hence, the posterior inference in BNNs can hardly reduce to MLE, leading to the performance divergence between BNNs and deterministic NNs. Empirically, in general, we deploy BNNs to seek for uncertainty estimation. On top of this, we want the model to be accurate. Therefore, it does not make sense to adopt “very flat priors and highly confident variational distributions” to achieve high performance while discarding the estimation of predictive uncertainty.
>
>
> Q7: References.
> A: Thanks for the suggestions. We checked these works and found that they are not so closely related to DBSN, which performs Bayesian structure learning with variational inference techniques. Anyway, we cited them in the related work as general Bayesian deep learning methods.
>
> [1] Georgi Dikov and Justin Bayer. Bayesian learning of neural network architectures. AISTATS 2019.
> [2] Hanxiao Liu, Karen Simonyan, and Yiming Yang. DARTS: Differentiable architecture search. ICLR 2019.
> [3] Sirui Xie, Hehui Zheng, Chunxiao Liu, and Liang Lin. SNAS: stochastic neural architecture
> search. ICLR 2019.
> [4] William Bialek, Ilya Nemenman and Naftali Tishby. Predictability, Complexity, and Learning, Neural Computation 13, 2409–2463, 2001.

---

> ### Author Response · Authors · 2019-11-13
> **Thank you for the feedback! (Part 1)**
>
> We thank the reviewer for the comments, though there may be quite a few misunderstandings. We clarify the potential misunderstandings and address the detailed concerns below.
>
>
> Q1: About the prior: “the priors of $\alpha$ are parameterized the same as the variational distributions thus the KL term is effectively zero”:
> A: This is indeed a misunderstanding. As stated in the beginning of Section 3 and the Setup part of Section 5.1, the priors are set to be concrete distributions with uniform class probabilities. Namely, the parameter $\theta_0$ in the prior $p(\alpha | \theta_0)$ are uniform values, different from the variational parameters $\theta$ of $q(\alpha | \theta)$, thus the KL is not zero. By optimizing the negative ELBO in Eq.(4), the variational distribution $q(\alpha | \theta)$ considers both the prior and the data likelihood; essentially it approximates the true model posterior $p(\alpha| D, w)$. We made this clearer in revision.
>
>
> Q2: About the performance of DARTS and re-training:
> A: First, we clarify that DBSN does not employ a re-training stage: DBSN is in a Bayesian framework instead of meta-learning. This was explicitly pointed out in the last parts of Section 2.2&3.3. Regarding the DARTS baseline, note that the results in DARTS paper are not directly comparable. We have re-implemented the first-order algorithm of DARTS using the same structure learning space as DBSN (stated in the Baseline part of Section 5.1), and test the trained super network directly, without an extra re-training stage. This DARTS baseline has the same structure learning space, # of parameters, and training process as DBSN, thus it is definitely fair to compare DBSN with it. Empirically, DARTS is much worse than DBSN as it only trains the weights on half of the training set without re-training, and this is also explained in the last part of Section 5.1. We also clarify that DARTS is not a main baseline of DBSN: in the aspect of predictive performance, we mainly compare with ResNet and DenseNet; in the aspect of uncertainty, which DARTS is lack of, we compare DBSN to NEK-FAC, Random $\alpha$, Dropout, and Drop-path. The comparison between DBSN and DARTS is only expected to show that DBSN is an appealing choice for effective neural structure learning with only one-stage training.
>
>
> Q3: Comparison with Dikov & Bayer 2019:
> A: We emphasize that the major contribution of DBSN compared to Dikov & Bayer 2019 is that we “explore an under-explored area at the intersection of state-of-the-art network architecture search algorithms and Bayesian neural networks”, as appreciated by Reviewer 1. The difference lies not only in the scalability, but also in the definition of structure space. On one hand, the effectiveness of our adopted DARTS-like structure space is well evaluated in previous works (Liu et al., 2019; Xie et al., 2019), which implicitly reveals the superiority of DBSN over Dikov & Bayer 2019. On the other hand, one can simply imagine that the structure space considered by Dikov & Bayer 2019, i.e. the layer size and network depth, is notably more restrictive than that of DBSN. For example, dropping one layer in Dikov & Bayer 2019 equals to dropping all the redundant connections from this layer’s predecessors to this layer in DBSN. Technically, we also propose novel strategies to perform more feasible and efficient training (Section 3.1 and 3.2), which further augments our contributions. Anyway, we have been working on implementing the structure distribution of Dikov & Bayer 2019, and we will try to provide explicit comparisons in the latter version.
>
>
> Q4: About the comment on “the authors mentioned in the introduction …; sample different architecture from the alpha learned in DARTS and DBSN, train several networks, ensemble them...”:
> A: We kindly speculate that there are still misunderstandings of the reviewer in this part. First, we clarify that we did not claim “DBSN brings more calibrated uncertainty comparing to ensembling different architectures”. Instead, we claim that DBSN “yields more diverse predictions” in comparison to the BNNs with weight uncertainty, and this is also explicitly expressed in the original Section 3.3. Our claim is powerfully supported by the ECE experiment results in Table 3, where we use $100$ MC samples to estimate the predictive distributions for DBSN, Dropout, NEK-FAC, etc. Moreover, the claim mentioned by R2 can hardly be true. If a bunch of networks with different architectures are individually trained with their own weights, they can naturally demonstrate improved expressive power than DBSN because they require much more parameters and more training time. The different sampled structures in DBSN share the same set of network weights under a principled Bayesian formulation. Therefore, DBSN is much more economical and time-saving than this kind of “explicit” ensembling, especially when the ensembling number is huge.

---

### Official Review · AnonReviewer3 · 2019-10-22
**Official Blind Review #3**

**Rating:** 3

**Review:**

This paper proposes to do approximate Bayesian inference in neural networks by treating the neural network structure as a random variable (RV), while inferring the parameters with point estimates.
While performing Bayesian inference for the neural network structure is sensible, I am not convinced by the approach taken in this work.

The biggest problem is that the model uses a point estimate of the same weights for different, random network structures.
Major problems:
- In the motivation the authors write “DBSN places distributions on the network structure, introducing more global randomness, thus is probable to yield more diverse predictions, and ensembling them brings more calibrated uncertainty”. What is “more global randomness”? This is used multiple times. Does it refer to the hierarchy in the graphical model? Please be precise here and point that out in your model by using an equation or graphical model. Or is it just an intuition?
- Generally, I would agree that integrating out multiple network structures provides better calibrated uncertainty. However, given that the authors use point estimates for the weights, it is not clear if that is still true, especially since the number of different architectures used in practice is small.
- What’s more, the approach uses *the same* point estimates for different structures. This leads to a graphical model, where the weights are not conditioned on the architecture/structure. This modeling choice could be a big limitation, because the weights now have to fit multiple different architectures; it may thus defeat the calibration completely. One can easily imagine that only a single random architecture works well with the learned point estimates, thus resulting in an (almost) deterministic model. I assume that this modeling choice was made for practical reasons, but could you expand on its implications / interpretation / limitations? Does the posterior of such a constrained model not quickly converge to an “almost” dirac, effectively just one network structure?
- Sec. 3.1. presents the above problem resulting from a modeling decision as a “training challenge”. To counter this problem, the authors propose to reduce the variance of the structure distribution. By doing so, the approach becomes even less Bayesian and the predictive uncertainty becomes even less reliable.
- “We only learn the connections between the B internal nodes, as shown in Appendix D”. All deterministic weights are learned, but the structure only for some parts of the model? If this is the case, then approach becomes again less probabilistic.

Regarding the experiments, the stddevs are calculated from 3(!) independent runs and thus completely misleading (imagine the stddev of the stddev estimate).

In summary, the model choice of point estimates for the weights, which are not conditioned on the architecture, leads to various problems. The authors have to introduce tricks such as reducing the variance of the random network structures or learning only a part of the whole structure to make the approach converge. The resulting probabilistic model and its predictive uncertainty is questionable. For this reason, this paper should be rejected.

Minor problems
- Sec. 3.2. “Improvements of the structure learning space”. What is a “structure learning space”?
- Section 3 introduces the ELBO in Eq. (4) before the complete model is specified. Please specify the whole model first. How do w and alpha depend on each other in your model?
- The prior for the weights is omitted; at the same time it is mentioned in the experiments (Sec.5.1.) that weight decay is applied. Why not just be explicit about it and say that a Gaussian prior is used?
- Background Sec. 2.2. is not clear.  what is a cell? some deterministic transformation in general? bunch of neural network layers? What are the operation (last term in Eq. (2)) doing? This is not detailed and abstract to me. Are the alphas probabilities? Is Eq. (2) consequently a mixture model of different architectures? Or is this here just a weighted sum, where the weights take arbitrary values? A small visualization (additionally) might help here, but can probably be rectified by better explanation.
- Bayesian reasoning on the structure. Inference?
- Writing that you propose a new “framework” is a bit grandiose for what is actually proposed. There has been previous work in which the architecture is inferred as well and these approaches would certainly be part of the same framework. Please just say model/algorithm/approach, whatever is applicable.
- new paragraph starting at “To empirically validate” in the intro.
- Before (4): “Then we rewrite the approximation error”. Eq. (4) is the ELBO, this is not an approximation error.


**Experience Assessment:**

I have read many papers in this area.

**Review Assessment: Checking Correctness Of Derivations And Theory:**

I assessed the sensibility of the derivations and theory.

**Review Assessment: Checking Correctness Of Experiments:**

I assessed the sensibility of the experiments.

**Review Assessment: Thoroughness In Paper Reading:**

I read the paper at least twice and used my best judgement in assessing the paper.

---

> ### Author Response · Authors · 2019-11-13
> **Thank you for the thorough feedback! (Part 3)**
>
> Q7: Concerns on minor problems:
> Thank you for your kind suggestions! We have updated the paper accordingly. Below are some selected responses:
>
> Q7.1: What is a “structure learning space”:
> A: The structure learning space means the support of the structure distribution. It is a set containing all the possible network structures, whose size grows exponentially with the number of paths with redundant operations.
>
> Q7.2: Model specification:
> A: We apologize for these points. We revised Section 3 and Fig. 1 to provide model specifications in detail. As shown in the updated Fig. 1, $w$ and $\alpha$ are independent variables.
>
> Q7.3: About weight decay on $w$:
> A: Thank you. We agree with the reviewer that the weight decay used to regularize $w$ essentially implies a Gaussian prior on the weights. Then, in practice, DBSN performs maximum a posteriori (MAP) estimation of $w$, namely, estimating the mode of $w$’s posterior distribution $p(w|D)$. Therefore, we can regard DBSN as doing an approximation to Bayesian inference on $w$, which is much more practical for the model’s training.
>
> Q7.4: Background Sec. 2.2. is not clear:
> A: We apologize for that. We updated the paper to make cell-based NAS clearer to understand. A cell is a network module containing several computational nodes, i.e. tensors, which can also be viewed as a bunch of layers. The last term $o^{(i,j)}_k(N^i; w)$ in Eq. (2) is the output through the $k$-th operation (e.g., convolution, skip connection, etc.) on $N^i$. The operation is equipped with a subset of $w$ as parameters, thus $w$ is added as a condition in the notation. As we have stated, the $\alpha^{(i,j)}$ are the gating weights of the different $K$ operations and are in a K-dimensional simplex. Regarding the visualization, we kindly remind that it has already been provided in Fig. 1.
>
> Q7.5: Other minor problems:
> A: We addressed them in the revised paper. We hope the new version is clearer and easier to follow. Feel free to let us know if there is still anything unclear.
>
>
> [1] Yarin Gal and Zoubin Ghahramani. Dropout as a bayesian approximation: Representing model uncertainty in deep learning. ICML 2016.
> [2] Yarin Gal, Jiri Hron, and Alex Kendall: Concrete Dropout. NIPS 2017.
> [3] Gao Huang, Yu Sun, Zhuang Liu, Daniel Sedra, and Kilian Q Weinberger. Deep networks with stochastic depth. ECCV 2016.
> [4] Matthew Mackay, Paul Vicol, Jonathan Lorraine, David Duvenaud, and Roger Grosse. Self-tuning networks: Bilevel optimization of hyperparameters using structured best-response functions. ICLR 2019.
> [5] Hanxiao Liu, Karen Simonyan, and Yiming Yang. DARTS: Differentiable architecture search. ICLR 2019.
> [6] Ziyu Wang, Tongzheng Ren, Jun Zhu, and Bo Zhang. Function space particle optimization for bayesian neural networks. ICLR 2019.

---

> ### Author Response · Authors · 2019-11-13
> **Thank you for the thorough feedback! (Part 2)**
>
> Q3: Concern on using point estimates for the weights:
> A: In theory, we can perform full Bayesian inference on both $w$ and $\alpha$, and we added such experiments (see the reply to R1’s second question as well as the updated Table 1&3). However, as stated in the introduction, there are still frustrating difficulties to achieve scalable Bayesian inference on the high-dimensional space of network weights, which is also proved by the results of the supplementary experiments: in Table 3 and Table 5, Fully Bayesian DBSN which is equipped with weight uncertainty demonstrates much worse calibration (i.e., ECE) than the counterpart DBSN. Hence, for now, we prefer to perform Bayesian inference on the low-dimensional network structure $\alpha$ while viewing the high-dimensional $w$ as point estimates, which is empirically testified to be feasible. Moreover, as pointed out by R3, the used weight decay in experiments implicitly imposes a Gaussian prior on $w$. Then, in practice, DBSN performs maximum a posteriori (MAP) estimation of $w$, namely, estimating the mode of $w$’s posterior distribution $p(w|D)$. Therefore, we can regard DBSN as doing an approximation to Bayesian inference on $w$, which is much more practical for the model’s training. We added these to Section 3.3. We also would like to push fully Bayesian DBSN one step further by introducing more flexible and more expressive variational distributions than the current factorized version for $w$. We leave this as future work.
>
>
> Q4: Concern on “variance reduction makes the approach even less Bayesian and the predictive uncertainty becomes even less reliable.”:
> A: Thanks. The existing results in Table 3, Fig. 4, and Fig. 5 give us direct comparisons between DBSN and DBSN* which does not use this variance reduction technique. We can see that DBSN achieves improved predictive uncertainty than DBSN*, providing strong empirical evidence to alleviate this concern.
>
> Technically, because of the weight sharing mechanism, the sampled structures during training may share some operations and weights (see this figure https://github.com/anonymousest/DBSN/blob/master/path.pdf ). Thus, there is indeed a challenge to train $w$ sufficiently well to be suitable for all the structures, because different structures may prefer different weights. The under-fitting of $w$ can bring bias in the learning of $\alpha$’s variational posterior. Thus, we adopted the variance reduction technique to help $w$ to fit the structure distribution better and eventually benefit the Bayesian structure learning. As stated in Section 3.1, we also take a necessary strategy (keep $\beta$ from being too small, i.e. at least 0.5) to avoid the structure distribution from degrading to a deterministic one. Note that the variance-reduced $q(\alpha)$ is still a valid and flexible variational distribution, which does not make the approach less Bayesian or less reliable.
>
>
> Q5: Concern on “We only learn the connections between …”:
> A: This statement means that we only build redundant operations between the internal nodes in the network cell, and the network structure is still defined as the selection of the redundant operations (in a weighted sum form). This design aims at easing structure learning and is similar to the common setup in differentiable NAS (Liu et al., 2019), where the backbone of the network is always fixed. Essentially, the Bayesian structure learning nature of DBSN is unchanged.
>
>
> Q6: Concern on “The stddevs calculated from 3 runs”:
> A: Thanks. We have conducted a set of 5 runs of DBSN on CIFAR-10, and the mean and stddev of the test error rate are 4.96 and 0.19 (%), respectively. Empirically, they are close to the corresponding results reported in Table 1, so the results in Table 1 are relatively plausible.

---

> ### Author Response · Authors · 2019-11-13
> **Thank you for the thorough feedback! (Part 1)**
>
> Thank you so much for your efforts in assessing our work. First, we clarify a potential misunderstanding, and then we answer the questions in detail.
>
>
> Q1: Clarification for a potential misunderstanding on “uses *the same* point estimates for different structures” and “converge to an “almost” dirac”:
> A: Indeed, the point estimate is a practical choice but it would not be a big limitation and the situation where the structure distribution converges to an “almost” dirac can hardly occur. Recall that DBSN builds a network with redundant operations (see Fig.1 and Section 2.2), thus $w$ is a set containing all the parameters of these operations. In DBSN, each sampled $\alpha$ corresponds to a specific network architecture. Once $\alpha$ (i.e., structure) is given, DBSN only forces the corresponding subset of $w$ to fit this structure faithfully. Conversely, different network structures adjust w.r.t. different subsets of $w$, and this effectively prevents the structure distribution from collapsing into an “almost” Dirac, which is R3’s concern. Note that the prior on $\alpha$, a concrete distribution with uniform class probabilities, also keeps the variational posterior away from converging to an “almost” dirac via the KL term (last term of Eq. (4)). Furthermore, our evaluation of DBSN on predictive uncertainty (see Table 3, Fig.4, and Fig.5) empirically verifies that the learned variational posterior is nondegenerative and well-calibrated.
>
> To better understand DBSN, we also added a graphical model of it (as well as the conventional BNNs for comparison) in the revision (See Fig. 1). $\alpha$ and $w$ are independent and factorized in our model. This kind of design is in the same spirit of weight-sharing NAS (Liu et al., 2019), which is proven to be a practical way for computationally efficient optimization. Though this is a useful contribution for practical Bayesian structure learning, we still admit that capturing the dependency of $w$ on $\alpha$ may be necessary for more accurate modeling. We leave this as future work.
>
> From another perspective, we emphasize that the widely used technique of MC Dropout can also be seen as an ensemble of different structures but with *the same* weights, belonging to the same model choice which concerns the reviewer. Extensive evaluations of MC Dropout (Gal & Ghahramani, 2016; Gal et al., 2017) have revealed that this kind of modeling choice is sensible and does not substantially introduce limitations on quantifying the uncertainty of real data. Therefore, the modeling choice is not a big limitation but a reasonable choice adopted by much previous work.
>
> We have updated the revision to make this clear. Please see Section 3.3.
>
>
> Q2: Concern on “more global randomness”:
> A: We apologize for the confusion. We have revised the paper to make it precise. It does not refer to the hierarchy in the graphical model (See Fig.1). In fact, here "randomness" primarily refers to the stochastic variations of the functions represented by deep neural networks. For both BNNs and our DBSN, the Bayesian treatment on either weights or structures essentially characterizes the uncertainty of functions. As compared to weights, network structures are more “global” in the sense that a tiny change of network structure yields a dramatic change in weights, e.g., adding a single connection in structure corresponds to introducing an extra dimension in the continuous weight space with an infinite number of possible values. Previous analysis (Wang et al., 2019) shows that due to the over-parameterization nature of BNNs, the state-of-the-art inference algorithms can suffer from mode collapsing, as multiple configurations of weights with a fixed structure correspond to one single function. In DBSN, by adopting the differentiable design of NAS, we can compactly represent the uncertainty of structures and perform inference in a much lower-dimensional space, to avoid the issues as suffered by the algorithms for BNNs. Our empirical results indeed proved the effectiveness of this approach.

---

### Official Review · AnonReviewer1 · 2019-10-23
**Official Blind Review #1**

**Rating:** 6

**Review:**

The paper combines ideas from neural architecture search (NAS) and Bayesian neural networks. Instead of maintaining uncertainty in network weights, the authors propose to retain uncertainty in the network structure. In particular, building on cell-based differentiable NAS, the authors infer a distribution over the gating weights of different cells incident onto a tensor while relying on point estimates for the weights inside each cell.

Overall, I liked the paper and vote for accepting it. The notion of maintaining uncertainty about the network structure is a sensible one, and the paper explores an as yet under-explored area at the intersection of state-of-the-art network architecture search algorithms and Bayesian neural networks. Moreover, this is accompanied by compelling empirics — results demonstrate gains in both predictive performance and calibration across diverse tasks and careful comparisons to sensible baselines are presented to evaluate various aspects of the proposed approach (Table 1).

Detailed Comments:
+ One issue the experiments fail to adequately disentangle is the effect of weight uncertainty vs structure uncertainty. Are the observed gains in accuracy and calibration simply a product of better structure learning? In particular, I would love to see a baseline where point estimates of \alpha are learned but posterior distribution over weights is inferred. I realize NEK-FAC was an attempt at providing such a comparison, but since it uses a different structure, it remains unclear whether it’s poor performance stems from the fundamental difficulty of learning posteriors over high dimensional weights or simply a sub-optimal network structure.

+ In a similar spirit, one can imagine a fully Bayesian DBSN where one infers posterior distributions overbite \alpha and w. Presumably, this would close the OOD entropy gap between random \alpha and DBSN.

+ How many Monte Carlo samples were used to evaluate Equation 8. In variational BNNs one often finds that using more MC samples doesn’t necessarily improve predictive accuracy over using the most likely sample (the mean if using a Gaussian variational family). It would be interesting to see predictive performance as a function of the number of MC samples for DBSN.

+ Clarity: While I am mostly upbeat about this paper, the writing could be significantly improved. While the overall ideas come across, there are several instances where the text appears muddled and needs a few more polishing passes.

**Experience Assessment:**

I have published one or two papers in this area.

**Review Assessment: Checking Correctness Of Derivations And Theory:**

I assessed the sensibility of the derivations and theory.

**Review Assessment: Checking Correctness Of Experiments:**

I assessed the sensibility of the experiments.

**Review Assessment: Thoroughness In Paper Reading:**

I read the paper at least twice and used my best judgement in assessing the paper.

---

> ### Author Response · Authors · 2019-11-13
> **Thank you for the feedback! (Part 2)**
>
> We realized the BBB method used for modeling weight uncertainty in BNN-LS & Fully Bayesian DBSN may be restrictive, resulting in such weakness. Therefore, we further implemented these two baselines with a most-recently proposed mean-field natural-gradient variational inference method, called Variational Online Gauss-Newton (VOGN) (Khan et al., 2018; Osawa et al., 2019). VOGN is known to work well with advanced techniques, e.g., momentum, batch normalisation, data augmentation. As claimed by Osawa et al. (2019), VOGN demonstrates comparable results to Adam. Then, we replaced the used BBB in BNN-LS and Fully Bayesian DBSN with VOGN, based on VOGN’s official repository (https://github.com/team-approx-bayes/dl-with-bayes ). With the original network size (B=7, 12 cells), the baselines trained with VOGN needed more than one hour for one epoch. Thus we adopted smaller networks (B=4, 3 cells), which have almost 41K parameters, for the two baselines. We also trained a DBSN in the same setting. The detailed parameters to initialize VOGN are here (https://github.com/anonymousest/DBSN/blob/master/dbsn/train_bnn_torchsso.py#L220 ). The experiments were conducted on CIFAR-10 and the results are provided in Table 5 of Appendix C. The predictive performance and uncertainty gaps between DBSN and the two baselines are very huge, which possibly results from the under-fitting of the high-dim weight distributions in BNN-LS and Fully Bayesian DBSN. We believe that our implementation is correct because our results are consistent with the original results in Table 1 of [5] (VOGN has 75.48% and 84.27% validation accuracy even with even larger 2.5M AlexNet and 11.1M ResNet-18 architectures). Further, DBSN is much more efficient than the two baselines. These comparisons strongly reveal the benefits of modeling structure uncertainty over modeling weight uncertainty, highlighting the practical value of DBSN.
>
> Q3: The number of Monte Carlo (MC) samples:
> A: Thanks for the suggestion. We plotted the changes of test loss, test error rate, and test ECE w.r.t. the number of MC samples used for testing DBSN in Fig. 6 (CIFAR-10) and Fig. 7 (CIFAR-100) in Appendix B. It is clear that ensembling the predictions from models with various sampled network structures improves the final predictive performance and calibration significantly. This is in marked contrast to the situation of classic variational BNNs, where using more MC samples does not necessarily bring improvement over using the most likely sample. As shown in the plots, we could better utilize 20+ MC samples to predict the unseen data, in order to adequately exploit the learned structure distribution. Indeed, we used 100 MC samples in all the experiments, except the adversarial attack experiments where we used 30 MC samples for attacking and evaluation.
>
> Q4: Writing:
> A: Thanks for the kind suggestions. We have tried our best to improve the writing in the revised version. Please feel free to comment if there are still misleading or confusing parts in the paper.
>
> [1] Charles Blundell, Julien Cornebise, Koray Kavukcuoglu, and Daan Wierstra. Weight uncertainty in neural network. ICML 2015.
> [2] Christos Louizos and Max Welling. Multiplicative normalizing flows for variational bayesian neural networks. ICML 2017.
> [3] Jiaxin Shi, Shengyang Sun, and Jun Zhu. Kernel implicit variational inference. ICLR 2018.
> [4] Mohammad Emtiyaz Khan, Didrik Nielsen, Voot Tangkaratt, Wu Lin, Yarin Gal, and Akash Srivas-tava.  Fast and scalable bayesian deep learning by weight-perturbation in adam.  ICML 2018.
> [5] Kazuki  Osawa,  Siddharth  Swaroop,  Anirudh  Jain,  Runa  Eschenhagen,  Richard  E  Turner,  RioYokota, and Mohammad Emtiyaz Khan.  Practical deep learning with Bayesian principles. NeurIPS 2019.

---

> ### Author Response · Authors · 2019-11-13
> **Thank you for the feedback! (Part 1)**
>
> We thank the reviewer for the positive review, which truly reflects many of the essential contributions of our work. We also appreciate the constructive suggestions for improving the presentation and evaluation. Regarding the specific comments, we answer as follows:
>
> Q1: Extra baseline to disentangle the effect of weight uncertainty vs structure uncertainty:
> A: Thanks for the valuable suggestions. In the revised version, we have implemented the baseline suggested by R1 (See Section 5.1 and Table 1 & 3). Concretely, based on the baseline PE (where $\alpha$ and $w$ are both point estimates), we replaced all the conv and fc layers with the Bayesian counterparts, namely, we placed uncertainty on $w$. We set $p(w):=\mathcal{N}(w|\mu_0,\Sigma_0)$ and $q(w):=\mathcal{N}(w|\mu,\Sigma)$ both to be fully-factorized Gaussian distributions, and minimized the objective $L(\alpha, \mu,\Sigma) = -E_{q(w|\mu,\Sigma)}[\log p(\mathcal{D}|\alpha,w)]+\lambda*D_{KL}(q(w|\mu,\Sigma)||p(w|\mu_0,\Sigma_0))$. We adopted the reparameterization trick for training, resembling Bayes by Backprop (BBB) (Blundell et al., 2015). We set the prior to be $\mathcal{N}(w|0, 0.1^2)$ and $\lambda$ to be 0.001. We denote this baseline by BNN-LS (Bayesian Neural Networks with Learnable Structure), and have updated Table 1 and Table 3. In classification tasks on CIFAR-10 and CIFAR-100, BNN-LS yields respectively 9.85% and 30.98% error rates, which are worse than those of DBSN. In Table 3, BNN-LS also reveals inferior calibration than DBSN, especially on CIFAR-10. We also noted that BNN-LS is slightly outperformed by PE in terms of predictive performance. Therefore, we can conclude that, due to the fundamental difficulties of modeling distributions over high dimensional weights, networks with weight uncertainty face challenges to achieve competitive performance, even with a learnable network structure. By contrast, modeling structure uncertainty is much easier and more practical and thus more likely to yield better performance.
>
> Q2: Fully Bayesian DBSN:
> A: Thanks for the suggestion. Indeed, a fully Bayesian neural network is desirable for its principled effectiveness for estimating uncertainty and avoiding over-fitting. However, as discussed in the Introduction and the response to Q1, performing probabilistic inference in a high-dimensional space of weights is highly challenging. Though much work has been done (See e.g., Louizos & Welling, 2017; Shi et al., 2018), it is still not very scalable yet. Our DBSN can be understood as an approximation to a fully Bayesian version, where we perform Bayesian inference on the low-dimensional space of structures ($\alpha \in \mathbb{R}^{84}$) while finding a point estimate (essentially a MAP estimate when using weight decay regularizer on weights) of the high-dimensional weights ($w \in \mathbb{R}^{1,000,000}$).
>
> Empirically, we added new results of a fully Bayesian version of DBSN (See Tables 1 & 3). Namely, we built a fully Bayesian DBSN by replacing all the conv and fc layers in DBSN with their Bayesian counterparts. This is analogous to what we did to implement BNN-LS, thus we follow all the settings of modeling $w$ in BNN-LS. We denote this baseline by Fully Bayesian DBSN and have added its results into Tables 1 & 3. We can see that the predictive performance and calibration of this baseline are not satisfying, owing to the difficulties of modeling weight uncertainty as stated above.

---

### Author Response · Authors · 2019-11-13
**Paper update overview**

We thank all the reviewers for their efforts in reviewing our paper and for providing constructive feedback. In the revised paper, we addressed the comments to strengthen our paper. In summary, here are the main changes that we made to the paper (highlighted in red in the paper):

- Added two baselines BNN-LS (BNNs with Learnable Structure) and Fully Bayesian DBSN, which both model weight uncertainty, in Table 1 & 3 and Appendix C (R1 and R3);
- Clarified that DBSN is performing approximate Bayesian inference in Section 3.3 (R1 and R3);
- Edited the experiment setup to clarify that we use 100 MC samples for test in all the experiments except the adversarial attack experiments (R1);
- Added new plots in Appendix B to demonstrate the effects of the number of MC samples in test phase (R1);
- Added explicit model specifications in Section 3, and added the probabilistic graphical model plots in Fig. 1 (R3, R2);
- Updated background section of cell-based NAS (R3);
- Clarified the weight sharing mechanism in Section 3.1 and Section 3.3 (R3);
- Addressed some minor problems (R3);
- Added references (R2);
- Updated the motivation of DBSN in the introduction.

We are open to more suggestions.

---

### Decision · Program_Chairs · 2019-12-19

**Decision:**

Reject

**Comment:**

The authors develop stochastic variational approaches to learn Bayesian "structure distributions" for neural networks. While the reviewers appreciated the updates to the paper made by the authors, there will still a number of remaining concerns. There were particularly concerns about the clarity of the paper (remarking on informality of language and lack of changes in the revision with respect to comments in the original review), and the fairness of comparisons. Regarding comparisons, one reviewer comments: "I do not agree that the comparison with DARTS is fair because the authors remove the options for retraining in both DARTS and DBSN. The reason DARTS trains using one half of the data and validate on the other is that it includes a retraining phase where all data is used. Therefore fair comparison should use the same procedure as DARTS (including a retraining phrase). At the very least, to compare methods without retraining, results of DARTS with more data (e.g., 80%) for training should be reported." The authors are encouraged to continue with this work, carefully accounting for reviewer comments in future revisions.